# Regulation of ABCA1 by AMD-Associated Genetic Variants and Hypoxia in iPSC-RPE

**DOI:** 10.3390/ijms23063194

**Published:** 2022-03-16

**Authors:** Florian Peters, Lynn J. A. Ebner, David Atac, Jordi Maggi, Wolfgang Berger, Anneke I. den Hollander, Christian Grimm

**Affiliations:** 1Laboratory for Retinal Cell Biology, Department of Ophthalmology, University Hospital Zurich, University of Zurich, 8952 Zurich, Switzerland; lynn.ebner@usz.ch; 2Institute of Medical Molecular Genetics, University of Zurich, 8952 Zurich, Switzerland; grubichatac@medmolgen.uzh.ch (D.A.); maggi@medmolgen.uzh.ch (J.M.); berger@medmolgen.uzh.ch (W.B.); 3Department of Ophthalmology, Radboud University Medical Center, 6525 Nijmegen, The Netherlands; anneke.denHollander@radboudumc.nl; 4AbbVie, Genomic Research Center, 200 Sidney Street, Cambridge, MA 02139, USA

**Keywords:** ABCA1, lipid accumulation, hypoxia, iPSC-RPE, reverse cholesterol transport, age-related macular degeneration, LXR agonist

## Abstract

Age-related macular degeneration (AMD) is a progressive disease of the macula characterized by atrophy of the retinal pigment epithelium (RPE) and photoreceptor degeneration, leading to severe vision loss at advanced stages in the elderly population. Impaired reverse cholesterol transport (RCT) as well as intracellular lipid accumulation in the RPE are implicated in AMD pathogenesis. Here, we focus on ATP-binding cassette transporter A1 (ABCA1), a major cholesterol transport protein in the RPE, and analyze conditions that lead to ABCA1 dysregulation in induced pluripotent stem cell (iPSC)-derived RPE cells (iRPEs). Our results indicate that the risk-conferring alleles rs1883025 (C) and rs2740488 (A) in *ABCA1* are associated with increased ABCA1 mRNA and protein levels and reduced efficiency of cholesterol efflux from the RPE. Hypoxia, an environmental risk factor for AMD, reduced expression of *ABCA1* and increased intracellular lipid accumulation. Treatment with a liver X receptor (LXR) agonist led to an increase in *ABCA1* expression and reduced lipid accumulation. Our data strengthen the homeostatic role of cholesterol efflux in the RPE and suggest that increasing cellular cholesterol export by stimulating *ABCA1* expression might lessen lipid load, improving RPE survival and reducing the risk of developing AMD.

## 1. Introduction

Age-related macular degeneration (AMD) is a progressive blinding disease which mainly affects the elderly population in western countries, with an estimated rising number of 288 million people affected by the year 2040 [1]. Patients with AMD lose their central and sharp vision due to degeneration of the light-sensitive photoreceptors in the central region of the retina, the macula. AMD can be divided into a “wet” and a “dry” form. Wet AMD is characterized by pathological neovascularization with vessels growing from the choroid into the retina, while the dry form emerges from atrophy of the retinal pigment epithelium (RPE) and subsequent photoreceptor degeneration. AMD is a complex and multifactorial disease and the molecular mechanisms of its pathogenesis are barely understood. However, there is emerging evidence that a disturbed lipid metabolism in the ageing eye might contribute to AMD development [2,3,4,5]. One hallmark of AMD is the accumulation of subretinal drusen, which are lipid–protein aggregates consisting of polar and neutral lipids, lipid-binding proteins, complement components and extracellular material [6,7]. Together with additional age-related changes, including thickening of Bruch’s membrane [8,9], reduced choroidal blood flow and choroidal ischemia [10,11], these deposits may establish a chronic hypoxic environment for the RPE and neuronal retina even before AMD develops. Thus, dysregulated lipid metabolism in the RPE may accelerate the generation of such a hypoxic environment and contribute to disease development, subsequently causing the degeneration of photoreceptor cells [12,13,14]. Furthermore, a major function of the RPE is the daily phagocytosis of photoreceptor outer segments (POSs) that are rich in lipids, including cholesterol [15,16]. In order to avoid an overload of excessive un-metabolized intracellular cholesterol, which is toxic for RPE in culture [17,18], RPE cells recycle cholesterol back to the photoreceptors or into the choroidal vasculature [19,20].

An important lipid transport protein is ATP-binding cassette transporter A1 (ABCA1), which flips lipids from the inner leaflet of the plasma membrane to the outer side and transfers them on extracellular lipid acceptor proteins, such as apolipoprotein AI (ApoAI), or nascent high-density lipoproteins (HDL). *ABCA1* expression is regulated by the liver X receptors (LXRs) α and β in a heterodimeric complex with retinoid X receptor (RXR) [21]. This transcription factor complex is activated by oxysterols in conditions of accumulated intracellular unesterified cholesterol to increase *ABCA1* expression and lipid export as a feedback response. Loss of *ABCA1* causes the rare Tangier disease, which leads to impaired cholesterol export, especially in macrophages, lack of HDL-cholesterol and development of atherosclerosis [22]. Recently, we developed a mouse model deficient for *Abca1* and its close relative *Abcg1* in the RPE [23]. ABCA1 and ABCG1 are both lipid export proteins that are capable of flipping phospholipids and cholesterol from the inner side of the cell membrane to the outer side. While ABCA1 can transfer these lipids to lipid acceptor proteins, such as ApoAI, to form nascent high-density lipoproteins (HDL), ABCG1 further enriches lipid load on such HDL particles [24]. Mice lacking *Abca1* and *Abcg1* in the RPE showed accumulation of lipids in and around the RPE with increasing age, which resulted in photoreceptor degeneration and reduced retinal function [23]. The same was observed in *Abca1* but not in *Abcg1* single knock-out mice, indicating that ABCA1 was the main driver of lipid accumulation in the sub-retinal space in mice.

The purpose of the current study was to identify conditions that reduce ABCA1 expression and function in RPE which might contribute to sub-retinal lipid accumulation and AMD pathogenesis in the ageing eye. Genome-wide association studies (GWAS) identified two single nucleotide polymorphisms (SNPs) in *ABCA1* (rs1883025 and rs2740488) that are in high linkage disequilibrium (r^2^ = 0.941) and are associated with AMD risk [25,26,27,28,29]. The major alleles of rs1883025 (C) and rs2740488 (A) are associated with an increased risk for AMD, while the minor alleles of rs1883025 (T) and rs2740488 (C) are associated with decreased AMD risk. However, the effect of these SNPs on ABCA1 expression or function in the RPE, and thus on lipid metabolism, is not yet known, since they localize to a non-coding region and are therefore not expected to alter the amino acid composition of the protein. Both SNPs are located in intron 2 in close proximity to each other and may potentially affect gene transcription, RNA splicing or RNA stability. A first indication of this possibility was provided by a previous study, where reduced ABCA1 expression both at the mRNA and protein levels was observed in lymphoblastoid cell lines of healthy individuals homozygous for the alleles associated with increased risk to develop AMD [23]. A second condition potentially affecting ABCA1 expression might arise due to the development of a hypoxic tissue environment in the ageing eye. Hypoxia may increase during ageing in retinal tissue and can be caused by thickening of Bruch’s membrane [8,9], drusen deposits [14,30] and/or reduced choroidal blood flow [10]. These conditions may reduce oxygen delivery and diffusion through the tissue and induce a cellular response, including dysregulation of ABCA1, which has been shown to react to hypoxia [31,32]. In this study, we used human iPSC-derived RPE (iRPE) cells to investigate the influence of these genetic and environmental conditions on ABCA1 expression and function which might be associated with the health status of RPE through intracellular lipid accumulation and, in turn, lead to retinal degeneration and vision impairment.

## 2. Results

### 2.1. ABCA1-Deficient iRPE Cells Have Reduced Cholesterol Efflux and Increased Intracellular Lipid Accumulation

To study the RPE-specific expression and function of ABCA1, we differentiated RPE cells from iPSCs (Gibco Episomal hiPSC Line, A18945). Differentiated iRPE cells exhibited the typical hexagonal morphology and formed tight junctions shown by immunofluorescence staining for ZO-1 (Figure 1A). Furthermore, cells developed pigmentation (Figure 1B) and expressed typical RPE marker genes at high levels (Ct values around 20), while expression of these genes was barely detectable (high Ct values > 33 (except for *OTX2*)) in the parental iPSC line (Figure 1C).

As proof of concept and to investigate the impact of the complete absence of *ABCA1* in human RPE cells, *ABCA1* knockout iPSCs were generated using CRISPR/Cas9 technologies and gRNAs targeting the *ABCA1* coding region either in exon 14 (gRNA with the highest on-target effect) or in exon 46 (harboring serine at position 2054, which is important for cholesterol efflux [33]). Disruption of the *ABCA1* sequence was validated by Sanger sequencing, which revealed a two-base pair insertion (TT) in exon 14 and a 5 bp deletion in exon 46, respectively. Both alterations lead to frame shifts in the *ABCA1* coding sequence (Figure 1D). *ABCA1* knockout iPSCs were differentiated to iRPE cells and no or very weak residual signals for ABCA1 protein were detected in Western blot analysis, confirming the successful gene disruption (Figure 1E). Of note, *ABCA1* knockout iRPEs had significantly reduced *ABCA1* mRNA levels compared to the parental unedited cells, indicating potential instabilities of the mutant *ABCA1* mRNAs (Figure 1F). On the other hand, we detected significantly increased *ABCG1* mRNA levels, a close relative of ABCA1, which might be upregulated to compensate for the loss of *ABCA1* (Figure 1G). We measured reverse cholesterol transport from parental and *ABCA1* knockout iRPEs, which were loaded with fluorescent-labeled cholesterol. Cholesterol efflux above the threshold was only detected in the parental cells in the presence of ApoAI in the cell culture medium and not in *ABCA1* knockout iRPEs or in the absence of a lipid acceptor (Figure 1H). Since *ABCA1* transcription is regulated by a heterodimeric complex consisting of LXR and retinoid X receptor (RXR) among other transcription factors, we examined whether the cholesterol efflux could be improved by applying the LXR agonist T0901317 to the iRPEs. Significantly increased efflux was detected in the parental cells but not in *ABCA1* knockout cells. We also tested whether loss of *ABCA1* leads to intracellular lipid accumulation as a consequence of reduced cholesterol export, which was previously reported for mouse RPE or macrophages [23,34]. By Nile red staining, a dye that binds to neutral lipids, such as cholesterol esters, we measured significantly increased intracellular lipids in the *ABCA1* knockout lines compared to the parental line (Figure 1I). Collectively, these results confirmed the successful knockout of *ABCA1* and demonstrated reduced cholesterol export from iRPEs to ApoAI, leading to intracellular lipid accumulation in the absence of ABCA1. This indicates that human RPE, similar to mouse RPE [23], depends on ABCA1 for cholesterol export and that ABCG1 or another mechanism cannot completely compensate for the absence or reduction of ABCA1 activity.

### 2.2. Polymorphisms in ABCA1 Are Associated with ABCA1 Expression and Cholesterol Efflux

Intracellular lipid accumulation is toxic for cells and the formation of lipid-loaded drusen is a hallmark of AMD [6,17,35]. GWAS identified polymorphisms in genes related to the cholesterol pathway that are associated with AMD risk. As *ABCA1* is one of the identified SNP-carrying genes, we aimed to determine whether the two SNPs rs1883025 and rs2740488 in intron 2 of *ABCA1* alter its expression and/or function in RPE cells, as a potential mechanism contributing to AMD pathogenesis. We generated iPSCs from six unrelated AMD patients. Three iPSC lines were homozygous for the *ABCA1* alleles associated with increased AMD risk and three lines were homozygous for the *ABCA1* alleles associated with decreased risk for AMD development (Figure 2A and Table 1). Sequencing for other AMD risk-associated SNPs revealed a heterogeneous distribution of these SNPs in all cell lines (Table 1). The iPSCs were differentiated to iRPEs, genotyped for both SNPs in *ABCA1* by Sanger sequencing and analyzed for expression of RPE marker genes (Appendix A). ABCA1 expression was analyzed both at the mRNA and protein levels. To our surprise, we measured significantly enhanced *ABCA1* mRNA and protein expression in the patient-derived iRPE lines harboring the increased risk alleles (Figure 2B,C). This difference was not detected in the undifferentiated iPSC lines (Appendix A). In contrast, expression of *ABCG1* and *NR1H3* (encoding LXRα) did not show alterations in iRPEs (Figure 2D,E) and iPSCs (Appendix A). Since variants in non-coding regions might affect gene expression via alternative splicing of the mRNA, we were interested to see whether rs1883025 and rs2740488 in intron 2 would alter the length of the *ABCA1* mRNA. However, using primer pairs binding in exon 2 and exon 3, we did not observe other PCR products in the iRPE cells of AMD patients carrying either increased or decreased AMD risk alleles in *ABCA1*, besides the expected 105 bp product resulting from normal splicing of the pre-mRNA (Appendix A). Next, we determined the functionality of ABCA1 in the patient iRPE lines by investigating reverse cholesterol transport. Despite having reduced ABCA1 expression, cells carrying the decreased AMD risk alleles in *ABCA1* showed a significantly increased cholesterol export (Figure 2F). An opposite but not significant trend was detected when cells were fed with BODIPY-cholesterol (fluorescence-tagged cholesterol)-labeled POSs, which were equally bound and phagocytosed by the cells (Figure 2G–I).

Next, we tested whether the stimulation of cells with the LXR agonist, mimicking a situation with high intracellular cholesterol load, might affect the expression and activity pattern of ABCA1. Both ABCA1 expression and cholesterol export were highly increased using the LXR agonist compared to untreated cells irrespective of the genetic background (Figure 2 and Figure 3). Although cells containing the increased risk genotype still expressed higher mRNA levels for ABCA1, protein levels were no longer significantly different (Figure 3A,B). Furthermore, LXR agonist-treatment enhanced cholesterol efflux capacity in all iRPE cells, resulting in non-significant differences between the two genotype groups after direct cell labeling or POS feeding, suggesting that the application of LXR agonists may be able to correct the cholesterol export deficiency of cells harboring the increased risk *ABCA1* genotype (Figure 2 and Figure 3C,D).

### 2.3. ABCA1 Expression and Function Is Regulated by Hypoxia in RPE

Common age-related changes in the eye are thickening of Bruch’s membrane, drusen deposits and reduced choroidal blood flow, leading to reduced oxygen supply to the retina, which induces local hypoxia with potentially detrimental outcomes for the metabolism of retinal cells. Since the *ABCA1* promoter contains potential hypoxia-inducible factor 1 (HIF1) binding elements [32] and hypoxia can affect expression of *ABCA1* and reverse cholesterol transport in macrophages [31], we aimed to determine whether hypoxia affects ABCA1 expression and function also in human RPE. Gene expression was analyzed in iRPEs cultivated in 4% oxygen for 3 days or normoxic (21% oxygen) environment. Although not reaching statistical significance, *ABCA1* mRNA levels were decreased by about 40% in cells with the decreased risk *ABCA1* genotypes and unaltered (with high standard deviation in the cells with the increased risk *ABCA1* genotypes) under hypoxic conditions (Figure 4A). A comparable tendency of hypoxic downregulation was observed for sterol regulatory element-binding protein 1c (*SREBP-1C*) and *NR1H3*, which encodes the oxysterols receptor LXR-alpha, an important transcription factor for the expression of *ABCA1* and *SREBP-1C*. This supports a general expression change of genes involved in lipid metabolism (Figure 4B,C). As proof of the applied hypoxia, we also analyzed expression of two hypoxia-associated genes, adrenomedullin (*ADM*) and 3-phosphoinositide-dependent protein kinase 1 (*PDK1*), which are regulated by HIF1. While *ADM* was strongly increased in both groups, *PDK1* was only minimally altered after three days in hypoxia, likely because it is upregulated only early and transiently in response to hypoxia [36] (Figure 4D,E). As a consequence of reduced *ABCA1* expression under hypoxia, iRPEs with the decreased but not with the increased risk *ABCA1* genotypes exported less cholesterol than under normoxia (Figure 4F).

The results for the patient iRPE cells prompted us to test the regulation of ABCA1 in RPE cells under hypoxia in more detail. We isolated and analyzed mRNA from commercial iRPEs (Figure 1) after 3 days in 4% oxygen. While *NR1H3* showed significantly increased expression after 24 h, *ABCA1* expression was only slightly increased (Figure 5A). However, analogous to the patient cell lines, both *ABCA1* and *NR1H3* decreased to 25% and 40%, respectively, on day 3 compared to the normoxic condition. Expression of the HIF1 target genes *ADM* and *PDK1* increased significantly in the hypoxic environment, but levels of *PDK1*, as an early response gene, declined again after 24 h, which we also observed in the patient-derived cells (Figure 4D and Figure 5B,C). To address whether downregulation of *ABCA1* in hypoxic iRPE cells is a general phenomenon for the RPE or specific to human cells, we analyzed levels of mouse *Abca1* in the RPE/choroid of mice that had been exposed to 14% O_2_ for 11 weeks [37]. Strikingly, we found *Abca1* to be significantly downregulated under chronic hypoxia in vivo, which is in line with our in vitro finding for human iRPEs (Figure 5D). In accordance with the reduced expression of *ABCA1* in hypoxic iRPE cells, cholesterol efflux was reduced by about 10% under hypoxia compared to the normoxic environment. Treatment with the LXR agonist drastically stimulated cholesterol efflux in hypoxia (Figure 5E). In line with previous findings [38], intracellular lipid accumulation was highly increased in iRPEs under hypoxic conditions (Figure 5F,G). Since LXR agonist-treatment enhanced ABCA1-mediated cholesterol efflux under hypoxia, we investigated whether stimulation of *ABCA1* transcription can decrease intracellular lipid accumulation. Indeed, we observed significantly less lipid accumulation in cells treated with the LXR agonist. Collectively, our results show that ABCA1 expression and function is reduced under chronic hypoxia both in vivo and in vitro and that stimulation of *ABCA1* transcription by an LXR agonist ameliorates cholesterol efflux in RPE cells under these conditions.

## 3. Discussion

Accumulation of lipid deposits in peripheral tissues, which is toxic to cells and can cause local hypoxic environments, is a hallmark of several age-related diseases, including AMD [6,7]. However, the exact causes or mechanisms that lead to the reduced transport of lipids for clearance are not completely understood. One possibility is the effect of variants in cholesterol transport-associated genes, such as *ABCA1*, *LIPC*, *APOE* and *CETP*, which were identified in several GWAS studies [25,27,29]. Depending on the localization and nature of the polymorphisms, they might alter the amino acid sequence affecting folding and/or function of the respective protein or influence gene expression on a transcriptional and/or post-transcriptional level [3].

In this study, we analyzed the regulation and function of the cholesterol transporter ABCA1 in iRPE cells generated from patients harboring *ABCA1* genotypes that have been associated either with increased or decreased risk to develop AMD. Furthermore, we identified hypoxia as a causative environmental condition to downregulate *ABCA1* in RPE both in vivo and in vitro, leading to diminished lipid efflux and intracellular lipid accumulation, consequences that may promote development of AMD. A prominent role for ABCA1 has already been described for atherosclerosis [31,39], where *ABCA1* deficiency in macrophages that drive foam cell formation leads to the formation of atherosclerotic plaques. Furthermore, biallelic mutations in the coding region of *ABCA1* cause the very rare Tangier disease characterized by a virtual absence of plasma HDL and ApoAI and the presence of cholesterol-laden macrophages in various tissues, including tonsils, spleen and bone marrow [40]. Even though no ocular phenotype besides corneal opacity in some patients has been described in Tangier disease to date [41], a functional cholesterol metabolism is crucial for normal RPE and photoreceptor homeostasis. Reverse cholesterol transport via ABCA1 in mouse RPE was recently shown to be necessary to prevent intracellular lipid accumulation, RPE atrophy and subsequent photoreceptor degeneration in mice [23]. In the present study, we show that lack of *ABCA1* impairs cholesterol efflux and increases intracellular neutral lipid load also in fully differentiated human iRPEs. In contrast to macrophages [42], ABCG1 does not appear to contribute significantly to reverse cholesterol transport in mouse [23] and human (Figure 1H) RPE cells and cannot compensate for ABCA1 deficiency. In contrast, ABCA1 is able to compensate for the lack of ABCG1 activity, as seen in *Abcg1*^−/−^ macrophages [43] and *Abcg1*-deficient mouse RPE [23], indicating that ABCA1 might be the most relevant cholesterol exporter in the RPE. Nevertheless, we cannot completely exclude a partial compensation of ABCA1 deficiency by ABCG1. ABCG1-mediated efflux was not measured when using ApoAI as a lipid acceptor and we did not analyze basal efflux due to the reduced assay sensitivity using BODIPY-cholesterol.

Generating RPE cells through the differentiation of iPSCs of AMD patients harboring gene polymorphisms associated with either an increased or decreased risk for AMD allows the study of the impact of genetic variants on cellular function. The approach also allows the testing of potential treatments to improve the function or support of patient-derived cells. Several studies have focused on genes with a very strong association with AMD, including the *CFH* variant rs1061170 (Y402H) or *ARMS2* (rs10490924) and *HTRA1* (rs11200638). iRPEs expressing *CFH^Y402H^* mimicked key hallmarks of AMD, such as cellular stress, inflammation, impaired mitochondrial function and accumulation of lipid droplets [44,45]. *HTRA1/ARMS2* risk alleles led to increased expression of inflammatory and complement factors and higher susceptibility to oxidative damage in RPE [46,47]. To determine whether AMD-associated SNP genotypes in the non-coding region of *ABCA1* have an impact on ABCA1 expression and/or function, we generated iRPEs from patients harboring the high- and low-risk genotypes at this locus. Surprisingly, cells carrying the increased risk genotypes expressed ABCA1 at higher levels but showed decreased or equal cholesterol efflux under normal conditions, suggesting that ABCA1 protein levels do not necessarily correlate with cholesterol export efficiency. The molecular basis of this observation is not known but does not appear to rely on divergent ABCA1 proteins potentially generated through differential splicing due to the polymorphisms in the second intron. A decreased cholesterol efflux efficiency was only detected after direct labeling but not after feeding the cells with POSs. Since efflux after direct labeling does not depend on several additional intracellular processes, including phagocytosis and lysosomal degradation, which may influence the presentation of lipids to the transporter proteins, it may more directly reflect the activity of ABCA1. Interestingly, the decreased cholesterol efflux efficiency in the presence of the alleles for increased risk can be corrected by applying the LXR agonist to increase ABCA1 expression, pointing to a potential therapeutic approach to increase efflux and reduce lipid load in the RPE of patients. Although cells treated with the LXR agonist still expressed the *ABCA1* mRNA from the increased-risk cells at higher levels, protein levels and cholesterol export no longer differed between the cells with the different risk alleles. This implies that stimulated RPE cells might be saturated with ABCA1 protein, preventing higher ABCA1 levels either through protein degradation or reduced efficiency of mRNA translation.

Although our data indicate a slight reduction in cholesterol efflux efficiency in iRPE cells harboring the genotype for increased risk, the results need to be interpreted with care as the sample size is small and variations in the genetic and/or epigenetic background and/or health status of the donor patients may influence results. This notion may be supported by our earlier report demonstrating a significantly lower *ABCA1* mRNA expression for the increased risk genotype after LXR agonist stimulation in lymphoblastoid cell lines derived from non-AMD patients [23]. Although these data seem conflicting, we believe that using differentiated iRPEs represents the in vivo situation more exactly than does the use of lymphoblastoid cells. Furthermore, the iRPE cells were generated from donor AMD patients and not healthy individuals, which may thus reflect the patient situation more accurately. In addition, GWAS studies stated only a moderate effect on AMD risk (odds ratio 0.9) for the *ABCA1* SNPs, suggesting a rather small overall effect on the AMD disease process [25,26,48]. We also analyzed the genetic background of our cell lines regarding other, more prominent variants associated with higher AMD risk (*CFH* Y402H, *ARMS2* A69S, and *C3* R102G) or protection (*CFH* I62V and *CFHR3/1 del*) and obtained a heterogeneous distribution of high- and low-risk genotypes of these variants.

Since AMD development is not only affected by genetic but also by environmental risk factors, such as reduced oxygen availability, we investigated whether hypoxia influences ABCA1 expression and cholesterol efflux in iRPE cells and mouse RPE. Chronic hypoxic exposure indeed reduced expression of *ABCA1* in murine RPE/choroid as well as in all iRPE cell lines except one and abolished the difference in the expression levels of both genotype groups observed in normoxia. Hypoxia not only reduced expression of *ABCA1* but also of *NR1H3*, one of its transcriptional regulators. Expression of *NR1H3* paralleled expression of *ABCA1* indicating that this transcription factor might mainly be responsible for the regulation of *ABCA1* under hypoxia, while the genotypes of AMD-associated SNPs, rs1883025 and rs2740488, had only minor effects. Nevertheless, a direct regulation of *ABCA1* by HIFs, which were shown to influence *ABCA1* expression and cholesterol efflux in macrophages [31,32], or hypoxia-associated miRNAs (e.g., miR-20a-5p) [49,50] might also be possible in hypoxic iRPE cells. In our iRPE cells, reduced oxygen supply led to a slightly decreased cholesterol efflux but resulted in roughly 15-fold elevated intracellular lipid levels. Intracellular lipid accumulation under hypoxia is in line with data from a mouse model exploring metabolism in RPE cells with an activated hypoxic response [38]. Such RPE cells exhibited a metabolic shift from lipid oxidation to glycolysis, resulting not only in reduced glucose availability for photoreceptors with detrimental effects for their survival but also in an increase of intracellular lipids, which is harmful for RPE function and survival [17,51,52]. Therefore, reduction of intracellular lipotoxicity by enhancing lipid efflux is suggested to be a promising approach to delay AMD progression [17,35]. We show that stimulating *ABCA1* expression using an LXR agonist significantly increased cholesterol efflux (Figure 5E) under hypoxia and reduced intracellular lipid accumulation (Figure 5F), demonstrating that a treatment paradigm with an LXR agonist could be a promising approach. However, LXR agonist application did not completely prevent intracellular lipid accumulation in hypoxic RPE cells, indicating that a successful treatment protocol may need to target additional lipid metabolism-associated pathways to normalize intracellular lipid levels in hypoxia.

In conclusion, we demonstrated that iPSC-derived RPE cells with the *ABCA1* genotype associated with increased risk to develop AMD used in our study showed decreased cholesterol export efficiency and that hypoxia as an environmental risk factor reduced *ABCA1* expression and increased intracellular lipid accumulation. Our study supports the hypothesis that enhancing ABCA1-mediated efflux in RPE cells, e.g., by using an LXR agonist, could be a feasible treatment option to normalize efflux and intracellular accumulation of lipids in AMD patients.

## 4. Materials and Methods

### 4.1. Human Subject Recruitment and iPSC Generation

The study was approved by the local ethical committee at the Radboud University Medical Center and was performed in accordance with the tenets of the Declaration of Helsinki. Individuals, all of whom provided written informed consent, were selected from the European Genetic Database (EUGENDA), a large multicenter database for clinical and molecular analysis of AMD. The disease status of the individuals was determined by certified graders based on color fundus photographs, OCT and fluorescent angiography, if available [53]. iPSCs were generated at the Radboud Stem Cell Technology Center (Radboudumc, Nijmegen, Netherlands) from EBV-immortalized B-lymphocytes of three individuals who were homozygous for *ABCA1* genotypes conferring decreased risk for AMD (rs1883025 TT and rs2740488 CC) and from three individuals who were homozygous for *ABCA1* genotypes conferring increased risk of AMD (rs1883025 CC and rs2740488 AA) by nucleofection with episomal vectors containing the genes *OCT3/4*, *SOX2*, *KLF4*, *L-MYC* and *LIN28*. Quality and pluripotency of the generated iPSCs were tested by stem cell marker staining, stem cell gene expression and karyotyping (Appendix A). All iPSC cell lines were genotyped for other polymorphisms associated with higher risk of AMD development (Table 1).

### 4.2. Genotyping of iPSC Cell Lines for AMD-Associated Polymorphisms by Next-Generation Sequencing

Long-range PCRs were performed to amplify the genomic regions surrounding the SNPs rs1061170, rs10490924, rs2230199, rs800292, and rs12144939 for each cell line, followed by sequencing on a MiSeq instrument (Illumina, San Diego, CA, USA), as previously described [54]. Briefly, fragments of 17293, 11254 and 4766 bp were amplified with Takara LA Taq DNA polymerase (Takara Bio, Kasatsu, Japan) for the SNPs in *CFH*, *ARMS2* and *C3*, respectively. Primers are available upon request. The long-range PCR reaction mix contained 0.3 μL TaKara LA Taq, 3 μL 10X LA PCR Buffer II (Mg^2+^ plus), 3 μL 10X Solution S (Solis BioDyne, Tartu, Estonia), 4.8 μL dNTP Mixture, 1.2 μL forward primer (10 μM), 1.2 μL reverse primer (10 μM), 5 μL gDNA template (10 ng/μL) and 11.5 μL H_2_O.

PCR products were quantified with a Qubit dsDNA High Sensitivity kit (Thermo Fisher Scientific, Waltham, MA, USA) and diluted to 10 ng/μL. Diluted PCR products from the same cell line were pooled and fragmented with a Covaris M220 instrument (Covaris, Woburn, MA, USA) to a target size of 350–400 bp. Subsequently, the TruSeq DNA Nano kit (Illumina) was used to prepare the libraries according to the manufacturer’s instructions. The libraries were quantified with a Qubit dsDNA High Sensitivity kit and diluted to 4 nM. Denatured libraries were loaded on a MiSeq instrument at a final concentration of 12 pM. Secondary analysis of raw sequencing data was performed using software from the Genome Analysis Toolkit (GATK), Picard tools collection and the Burrows–Wheeler Aligner.

### 4.3. Differentiation of iPSCs to iRPE Cells

A control iPSC line was purchased (Gibco Episomal hiPSC Line, A18945, Thermo Fisher Scientific, Waltham, MA, USA) and maintained in TeSR-E8 medium (Stemcell Technologies, Vancouver, BC, Canada) on Matrigel-coated dishes (Growth factor reduced; Corning, Bedford, MA, USA) along the six patient-derived iPSC lines generated by the Radboud Stem Cell Technology Center (see above). Differentiation of iPSCs to RPE cells was performed as previously described [55,56]. iPSCs were seeded on Matrigel-coated dishes and grown to 70% confluency. Medium was changed to differentiation medium consisting of high-glucose DMEM (Sigma-Aldrich, St. Louis, MO, USA), 20% knockout serum replacement (Gibco, Thermo Fisher Scientific, Waltham, MA, USA), 1% NEAA (Sigma-Aldrich, St. Louis, MO, USA), 50 μM β-mercaptoethanol and 100 μg/mL Primocin (InvivoGen, San Diego, CA, USA). Differentiation medium was supplemented with 10 mM nicotinamide for one week, followed by supplementation with 100 ng/mL activin-A (Peprotech, Cranbury, NJ, USA) for another week and 3 μM CHIR99021 (Peprotech, Cranbury, NJ, USA) for 4 weeks. Six weeks after the start of differentiation, pigmented and RPE-shaped patches were manually picked or detached using TrypLE express (Gibco, Thermo Fisher Scientific, Waltham, MA, USA) and seeded on Matrigel-coated dishes and cultivated in FMN medium consisting of DMEM:F12 (Gibco, Thermo Fisher Scientific, Waltham, MA, USA), 1% NEAA (Sigma-Aldrich, St. Louis, MO, USA), 1% N2 Supplement (Gibco, Thermo Fisher Scientific, Waltham, MA, USA) and 100 μg/mL Primocin (InvivoGen, San Diego, CA, USA). iRPE cells were passaged once before banking at p2. Cells were thawed and seeded at p3 on Matrigel-coated dishes at 1.5 × 10^5^ cells/cm^2^ density and cultured for 4 weeks until the start of the experiments in FMN medium.

### 4.4. Generation of ABCA1 Knockout iPSCs by CRISPR-Cas9

*ABCA1* knockout in the Episomal hiPSC Line (A18945, Thermo Fisher Scientific, Waltham, MA, USA) was achieved by cloning specific gRNAs targeting exon 14 (5′-CGTACCGCATGTCCTCAAAG-3′) or exon 46 (5′-ATTTTTCTCCATACTTCACG-3′) of *ABCA1* into pSpCas9(BB)-2A-Puro (PX459) vectors (Addgene plasmid #48139, a gift from Feng Zhang) with an EF1α promoter sequence upstream of Cas9 and puroR. Potential genomic off-target sequences for the gRNA targeting exon 14 and the gRNA targeting exon 46 were detected only with four or three, respectively, mismatches and with a score of 1 or below (Appendix A). The two potential off-target sequences with the highest scores that fell within annotated genes were amplified and sequenced from the parental as well as the respective knockout lines. No sequence alterations were detected at these sites in the knockout lines (Appendix A).

For transfection, iPSCs were detached using TrypLE express (Gibco, Thermo Fisher Scientific, Waltham, MA, USA) and seeded as 0.5 × 10^5^ single cells into Matrigel-coated 24-well plates in TeSR-E8 medium (Stemcell Technologies, Vancouver, BC, Canada) supplemented with 1:100 RevitaCell (Thermo Fisher Scientific, Waltham, MA, USA). On the following day, cells were transfected using 1 μL Lipofectamine Stem (Thermo Fisher Scientific, Waltham, MA, USA) and 500 ng Cas9-sgRNA plasmids for 24 h. Transfected cells were selected by adding 0.5 μg/mL puromycin for the next 2 days to the medium. After the transfected cells recovered, they were seeded at low density into new Matrigel-coated dishes and expanded until they formed colonies of appropriate size. Single colonies were transferred manually into new dishes, expanded and harvested for DNA isolation. Non-homologous end joining (NHEJ) and sequence disruption in exon 14 or exon 46 was confirmed by Sanger sequencing.

### 4.5. PCR Amplification of ABCA1 SNPs and the Region Spanning Exon 2 and Exon 3

iRPEs were washed with PBS and lysed in Tissue Homogenization Buffer (500 mM KCl, 100 mM Tris-HCl, pH 8.3, 0.1 mg/mL gelatin, 0.45% IGEPAL CA-630, 0.45% Tween 20) with 50 μg/mL Proteinase K by scraping. Cell lysates were incubated at 55 °C for 1 h followed by 10 min denaturation at 95 °C. PCR reaction mix consisted of 0.2 μL Phusion HF Polymerase (New England Biolabs, Ipswich, MA, USA), 4 μL Phusion HF Reaction Buffer, 1.6 μL dNTPs (2.5 mM), 1 μL forward primer (10 μM), 1 μL reverse primer (10 μM) and 2 μL DNA, add 20 μL H_2_O. The following primer pairs were used for rs1883025 (forward: 5′-GAACCCTACCTGTGCTCCT-3′; reverse: 5′-TGTGCCAGAACTTGGCTTTA-3′); rs2740488 (forward: 5′-TAGAAGTGGGGAAAGGATGC-3′; reverse: 5′-GCTGGGATTATGGGCACA-3′). PCR products were loaded on agarose gels, extracted and analyzed by Sanger sequencing.

To analyze potential splice variants in *ABCA1* exon 2 and exon 3, PCR was performed as described above using cDNA from patient-derived iRPEs with increased or decreased risk polymorphisms for AMD and primer pairs binding in exon 2 and exon 3 of *ABCA1* (forward: 5′-TTGCTGCTGTGGAAGAACCTC-3′; reverse: 5′-CCGAACAGAGATCAGGATCAGG-3′). PCR products were separated on 1.5% agarose gel.

### 4.6. Immunofluorescence Staining

iRPE cells were grown on Matrigel-coated coverslips for 4 weeks. Cells were washed 2 times with PBS and fixed in 4% PFA for 15 min at RT, followed by blocking in 3% normal goat serum and 0.3% Triton x-100 in PBS for 1 h at RT. Primary antibody rabbit anti-ZO-1 (1:250, #40-2200c, Life Technologies, Thermo Fisher Scientific, Waltham, MA, USA) diluted in blocking buffer was applied overnight at 4 °C. Afterwards, coverslips were washed 3 times in PBS and incubated with secondary antibodies conjugated to Cy3 fluorophore (Jackson ImmunoResearch, Cambridgeshire, UK) for 1 h at RT in the dark. Coverslips were washed 3 times in PBS, incubated with 0.8 μg/mL DAPI in PBS, washed 3 times in PBS, once in ddH_2_O and mounted with mowiol. Immunofluorescence images were acquired using an Axio Imager K Z.2 microscope (Zeiss, Jena, Germany).

### 4.7. RNA Isolation, cDNA Synthesis and qPCR

RNA of iPSCs or iRPEs was isolated using an RNA Isolation Kit (Macherey-Nagel, Düren, Germany) according to the manufacturer’s instructions. cDNA was synthetized from equal amounts of RNA using oligo (dT) primers and M-MLV reverse transcriptase (Promega, Dübendorf, Switzerland). Semi-quantitative real-time PCR was performed using 10 ng of cDNA as template, PowerUp SYBR Green Master Mix (Thermo Fisher Scientific, Waltham, MA, USA) and primer pairs specific for the genes of interest (Table 2). Gene expression was normalized to *ACTB* or *RPL28*, as indicated in the figure legends.

### 4.8. Western Blot Analysis

iRPE cells were thawed at p3, seeded on Matrigel-coated plates at 1.5 × 10^5^ cells/cm^2^ and cultivated in FMN medium. One day before harvesting, cells were treated with 1 μM LXR agonist (T090137, Sigma-Aldrich, St. Louis, MO, USA) or DMSO overnight for 16 h. Cell lysates were prepared by direct lysis in RIPA buffer (50 mM Tris-HCl, pH 8.0, 150 mM NaCl, 0.1% SDS, 0.5% sodium deoxycholate, 1% Triton X-100, Protease Inhibitor Cocktail tablet (Roche, Basel, Switzerland)) and sonication. Protein concentration was determined using a BCA Assay Kit (Thermo Fisher Scientific, Waltham, MA, USA). Then, 50 μg protein lysates were separated by SDS-PAGE and transferred onto nitrocellulose membranes. Membranes were blocked in 5% non-fat milk in TBS-T and incubated with primary antibodies overnight at 4 °C: rabbit anti-ABCA1 (1:250, NB400-105, Novus Biologicals, Littleton, CO, USA), mouse anti-ACTB (1:10,000, A5441, Sigma-Aldrich, St. Louis, MO, USA). After washing, membranes were incubated with HRP-conjugated secondary antibodies at RT for 1 h and the signal developed using enhanced chemiluminescence (ECL) substrate (PerkinElmer, Schwerzenbach, Switzerland) and an X-ray developer. Band intensities were quantified using ImageJ and normalized to actin levels.

### 4.9. Cholesterol Efflux Assay

iRPE cells were thawed at passage 3 and 1 × 10^5^ cells were seeded on Matrigel-coated 24-well inserts and cultivated in FMN medium for 4 weeks. Cells were washed 2 times with PBS and incubated with 2 μM BODIPY-cholesterol (Cayman Chemical, Ann Arbor, MI, USA) diluted in phenolred-free FMN medium for 24 h. Afterwards, excess BODIPY-cholesterol was removed by two washes with PBS and cells were incubated in phenol red-free FMN containing 2 mg/mL BSA for 8 h. Cells were stimulated with 1 μM LXR agonist (T0901317, Sigma-Aldrich, St. Louis, MO, USA) or DMSO both in apical and basal media for 16 h overnight, followed by 2 washes with PBS to remove excess LXR agonist. ABCA1-mediated cholesterol efflux was achieved by adding 10 μg/mL ApoA1 (Peprotech, Cranbury, NJ, USA) into the apical medium consisting of phenol red-free FMN medium and 2 mg/mL BSA. After 6 h, apical and basal media were collected in separate tubes, centrifuged at 18,000× *g* for 10 min and stored in the dark. Cells were washed 3 times with PBS and lysed in 200 μL RIPA buffer for 30 min. Cell lysates were collected in tubes, centrifuged at 18,000× *g* for 10 min and stored in the dark. Then, 100 μL medium and cell lysate were transferred into dark 96-well plates and fluorescence was measured at 485/528 nm excitation/emission in a microplate reader (Synergy HT, BioTek, Agilent, Technologies, Santa Clara, CA, USA). Cholesterol efflux was calculated with the following equation: (apical medium counts × dilution factor)/[(apical medium counts × dilution factor) + (basal medium counts × dilution factor) + (cell lysate counts × dilution factor)] × 100.

For POS-mediated BODIPY-cholesterol labeling of cells, porcine POSs (prepared according to [57]) were washed 2 times with PBS and centrifuged at 3000× *g* for 3 min at RT. POSs were re-suspended in phenol red-free FMN medium supplemented with 2 mg/mL BSA and 10 μM BODIPY-cholesterol (Cayman Chemical, Ann Arbor, MI, USA) and incubated at 4 °C overnight on a shaker. The following day, POSs were washed 2 times with PBS, re-suspended in phenol red-free FMN medium with 2 mg/mL BSA and counted with a Neubauer chamber, after which 10 POSs/cell (approximately 3 × 10^6^ POSs per 24-well insert) were diluted in medium and transferred into 24-well inserts containing iRPE cells that were washed twice. After 8 h, medium was removed and cells were washed twice with PBS, followed by overnight stimulation with LXR agonist or DMSO and cholesterol efflux measurement, as described above.

### 4.10. POS Phagocytosis Assay

Porcine photoreceptor outer segments (POSs) were isolated as previously described and stored at −80 °C [57]. For phagocytosis assays, POSs were thawed on ice and washed twice with PBS and centrifuged at 3000× *g*. Around 10^7^ POSs were re-suspended in DMEM:F12 medium, supplemented with 10 μL FITC (1 mg/mL in DMSO; Sigma-Aldrich, St. Louis, MO, USA) and incubated overnight at 4 °C with mild agitation. The next day, POSs were washed twice with PBS, centrifuged at 3000× *g* and re-suspended in FMN medium. Ten POSs/cell were diluted in FMN medium and transferred into 24-wells with iRPEs grown on coverslips. After 4 h, excess POSs were removed by 3 washes with PBS. Cells were fixed in 4% PFA/PBS for 15 min at RT, followed by immunofluorescence staining for ZO-1 and nuclei counterstaining with DAPI. Fluorescence images were acquired using an Axio Imager K Z.2 microscope (Zeiss, Jena, Germany) and the numbers of bound and internalized POS particles and nuclei were quantified using ImageJ.

### 4.11. Nile Red Staining

Intracellular neutral lipids were stained by treating iRPEs with 2 μg/mL Nile red (Sigma-Aldrich, St. Louis, MO, USA) diluted in FMN medium for 30 min at 37 °C, as previously described [35]. Cells were washed 3× with PBS, fixed in 4% PFA for 10 min at RT, washed with PBS and mounted in mowiol. Fluorescence images were acquired with an Axio Imager K Z.2 microscope (Zeiss, Jena, Germany) and fluorescence intensities were calculated using ImageJ.

### 4.12. Animal Experiments

Animal experiments adhered to the ARVO Statement for the Use of Animals in Ophthalmic and Vision Research and the regulations of the Veterinary Authorities of Kanton Zurich, Switzerland (study approval reference number: ZH214/17). C57BL/6 mice were purchased from Charles River (Ecully, France) and kept in a 14:10 h light–dark cycle with lights on at six am and lights off at eight pm. Mice had access to food and water ad libitum. Chronic hypoxia was applied by keeping the mice under normobaric hypoxia (14% O_2_) for 11 weeks, followed by euthanasia and quick isolation of retina and RPE/choroid samples. Littermates under normoxic conditions served as a control group.

### 4.13. Statistical Analysis

Statistical analyses were carried out using GraphPad Prism 8. Statistical tests and the numbers of biological replicates are indicated in figure legends.

## Figures and Tables

**Figure 1 ijms-23-03194-f001:**
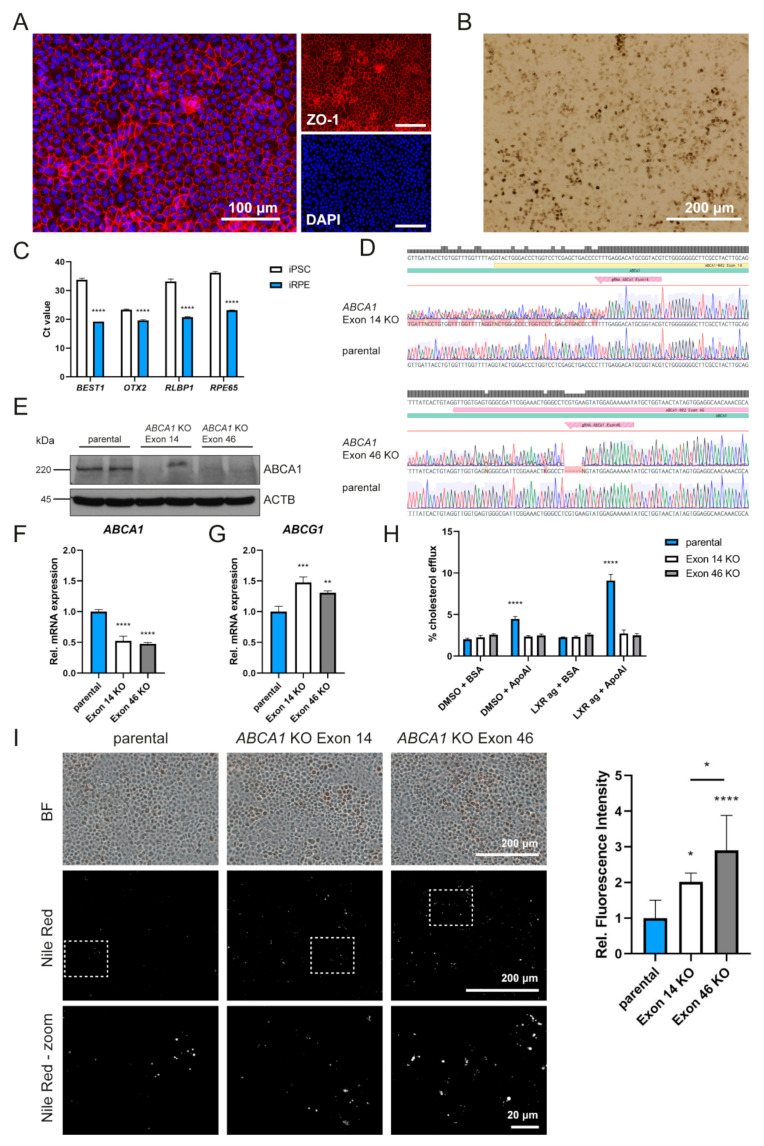
Generation, differentiation and analysis of *ABCA1*-deficient iRPEs. (**A**) Immunofluorescence staining for ZO-1 (red) in 4-week cultured iRPEs. Nuclei were stained with DAPI (blue). Scale bar = 100 μm. (**B**) Bright-field microscopy of iRPEs. (**C**) Ct values of RPE marker genes *BEST1*, *OTX2*, *RLBP1* and *RPE65* in iRPEs and parental iPSCs obtained by qPCR. Values shown are means ± SD (n = 3). Unpaired Student’s *t*-test. **** *p* < 0.0001. (**D**) Sequence alignment of parts of exon 14 (**top**) and exon 46 (**bottom**) of *ABCA1*-deficient iRPE cell clones and parental line. (**E**) Western blot of ABCA1 protein levels in *ABCA1*-deficient iRPE cell lines and parental cell line after 16 h of stimulation with 1 μM LXR agonist. Actin was detected as loading control. (**F**) Relative expression of *ABCA1* mRNA in *ABCA1*-deficient iRPE cell lines and parental cell line normalized to *ACTB*. Values shown are means ± SD (n = 3). One-way ANOVA with Tukey’s post hoc test. **** *p* < 0.0001. (**G**) Relative expression of *ABCG1* mRNA in *ABCA1*-deficient iRPE cell lines and parental cell line normalized to *ACTB*. Values shown are means ± SD (n = 3). One-way ANOVA with Tukey’s post hoc test. ** *p* < 0.01; *** *p* < 0.001. (**H**) Cholesterol efflux assay in iRPEs in the presence of ApoAI and/or LXR agonist (LXR ag). Values shown are means ± SD (n = 3). Two-way ANOVA with Bonferroni post hoc test vs. DMSO + BSA control. **** *p* < 0.0001. (**I**) Bright-field (BF) microscopy and Nile Red fluorescence microscopy (overview and close-up view marked with white squares) of 4-week cultured *ABCA1*-deficient iRPE cell lines and parental cell line. Relative fluorescence was quantified and is shown as mean ± SD (n = 8). One-way ANOVA with Tukey’s post hoc test. * *p* < 0.05, **** *p* < 0.0001.

**Figure 2 ijms-23-03194-f002:**
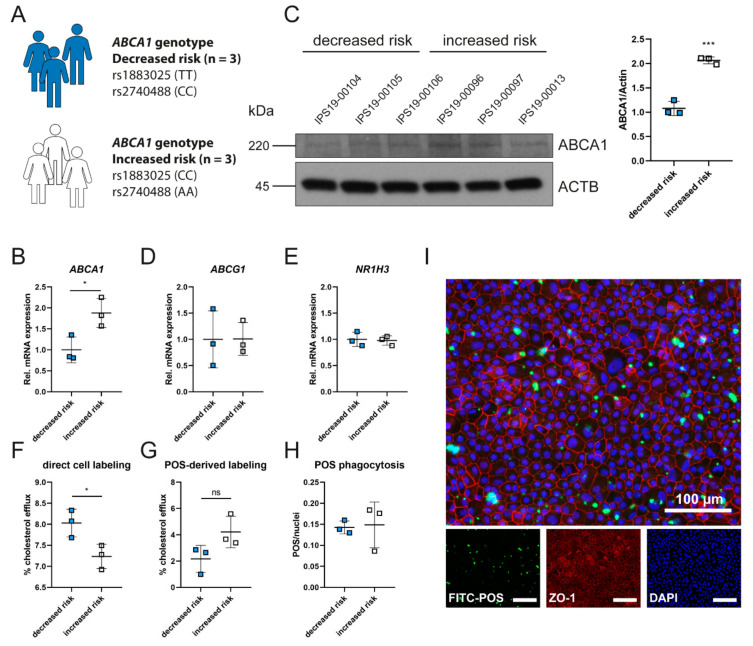
Basal ABCA1 expression and function in patient-derived iRPEs. (**A**) Representation of patient-derived iPSC lines and genotypes harboring polymorphisms in *ABCA1* that are associated with decreased or increased risk for AMD development. (**B**) Relative expression of *ABCA1* mRNA in patient-derived iRPEs under basal conditions normalized to *RPL28* and decreased risk group. (**C**) Western blot analysis of ABCA1 levels in patient-derived iRPEs under basal conditions. Actin was detected as loading control. ABCA1 expression was quantified and normalized to actin. (**D**,**E**) Relative expression of *ABCG1* (**D**) and *NR1H3* (**E**) mRNA in patient-derived iRPEs under basal conditions normalized to *RPL28* and decreased risk group. (**F**) Cholesterol efflux in patient-derived iRPEs after direct cell labeling and in the presence of ApoAI. (**G**) Cholesterol efflux in patient-derived iRPEs after phagocytosis of BODIPY-cholesterol-loaded POSs and in the presence of ApoAI. (**H**) Quantification of phagocytosed POSs per nuclei in patient-derived iRPEs. Values shown are means ± SD (n = 3). * *p* < 0.05; *** *p* < 0.001. Unpaired Student’s *t*-test. (**I**) Representative fluorescence microscopy images of phagocytosed FITC-labeled POSs (green) and staining for ZO-1 (red) and DAPI (blue) of increased risk cell line IPS19-00096. Scale bar = 100 μm.

**Figure 3 ijms-23-03194-f003:**
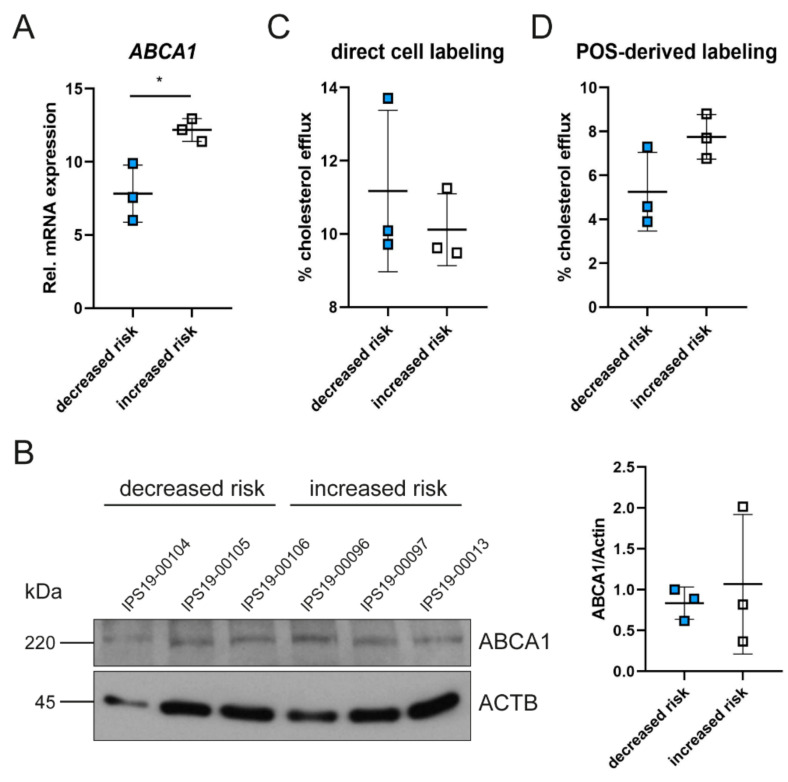
LXR agonist-stimulated ABCA1 expression and function in patient-derived iRPEs. Patient-derived iRPE lines were stimulated for 16 h with LXR agonist and are relative to unstimulated cells shown in Figure 2. (**A**) Expression of *ABCA1* mRNA normalized to *RPL28* and relative to unstimulated cells shown in Figure 2B. (**B**) Western blot analysis of ABCA1 protein levels. Actin was detected as loading control. ABCA1 expression was quantified and normalized to actin. (**C**) Cholesterol efflux assay after direct cell labeling and in the presence of ApoAI. (**D**) Cholesterol efflux assay after phagocytosis of BODIPY-cholesterol-loaded POSs and in the presence of ApoAI. Data are relative to unstimulated cells shown in Figure 2F,G, respectively. Data are presented as means ± SD (n = 3). Unpaired Student’s *t*-test. * *p* < 0.05.

**Figure 4 ijms-23-03194-f004:**
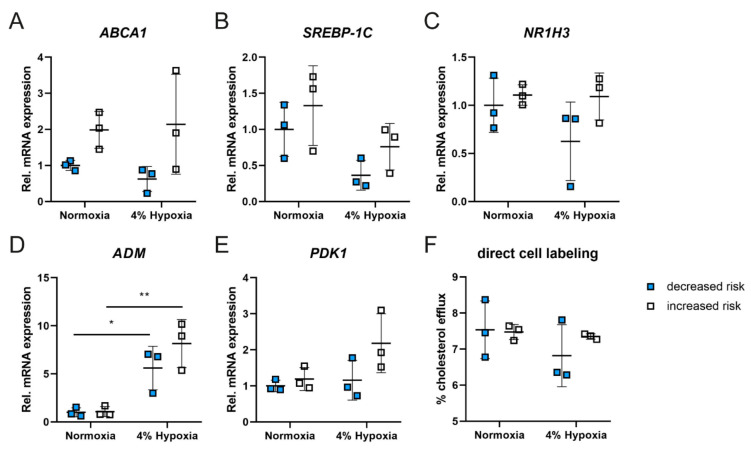
Gene expression and cholesterol efflux under hypoxia in patient-derived iRPEs. (**A**–**E**) Relative expression of (**A**) *ABCA1*, (**B**) *SREBP-1C*, (**C**) *NR1H3*, (**D**) *ADM* and (**E**) *PDK1* mRNA in patient-derived iRPEs after 3 days under 4% O_2_ (hypoxia) or 21% O_2_ (normoxia). All data were normalized to *RPL28* and are presented as means ± SD (n = 3). Two-way ANOVA with Tukey’s post hoc test. * *p* < 0.05; ** *p* < 0.01. (**F**) Cholesterol efflux assay in patient-derived iRPEs under 4% O_2_ (hypoxia) or 21% O_2_ (normoxia). Values shown are means ± SD (n = 3). Two-way ANOVA with Tuckey post hoc test.

**Figure 5 ijms-23-03194-f005:**
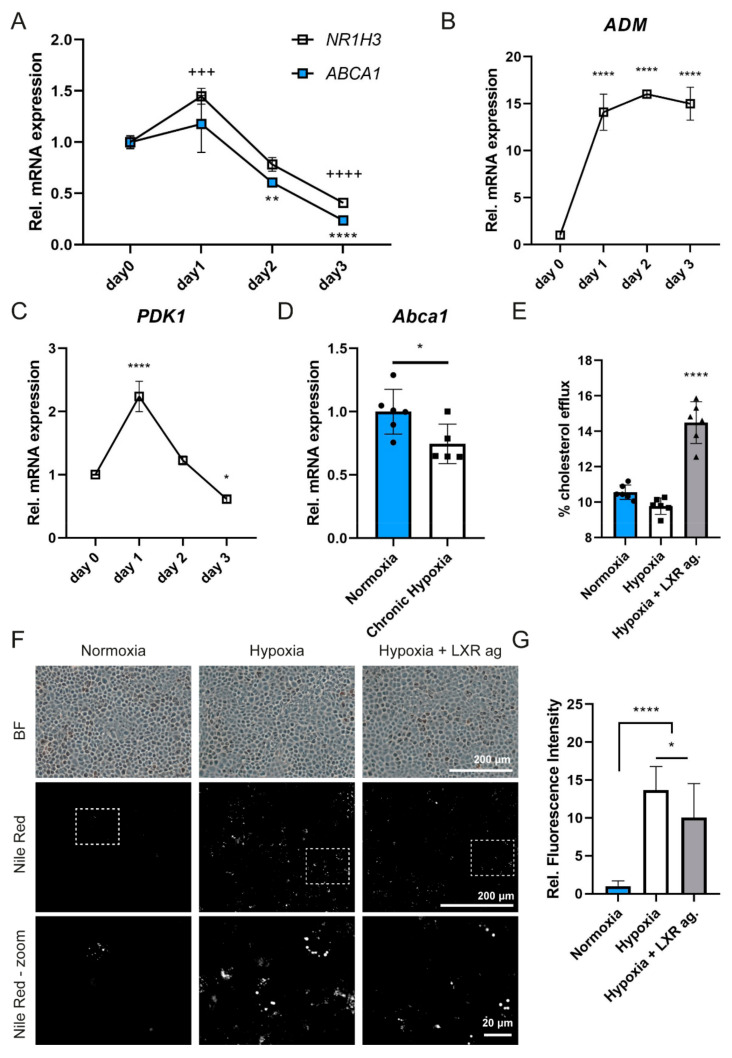
ABCA1 expression and function under hypoxia in iRPEs. (**A**–**C**) Relative expression of (**A**) *ABCA1* and *NR1H3*, (**B**) *ADM* and (**C**) *PDK1* mRNA in iRPEs before and after 1, 2 and 3 days incubation at 4% O_2_ (hypoxia). All data were normalized to *RPL28* and are presented as means ± SD (n = 3). One-way ANOVA with Tukey’s post hoc test to day 0. * *p* < 0.05; ** *p* < 0.01; ^+++^
*p* < 0.001; ****/^++++^
*p* < 0.0001. (**D**) Relative expression of *Abca1* in the RPE/choroid of mice kept for 11 weeks at 14% O_2_ (hypoxia) or in normoxic control conditions. Data are presented as means ± SD (n ≥ 5) and were normalized to *Actb*. Unpaired Student’s *t*-test. * *p* < 0.05. (**E**) Cholesterol efflux in the presence of ApoAI from normoxic or hypoxic (3 d at 4% O_2_) iRPEs after direct labeling and stimulation by LXR agonist as indicated. Data are presented as means ± SD (n = 6). One-way ANOVA with Tukey’s post hoc test. **** *p* < 0.0001. (**F**) Bright-field (BF) microscopy and Nile red fluorescence microscopy (overview and close-up view marked with white squares) of iRPEs after 3 days at normoxic or hypoxic (4% O_2_) conditions in the presence of ApoAI (all samples) and LXR agonist (as indicated). (**G**) Relative fluorescence of cells labeled with Nile red after 3 days of incubation in normoxic or hypoxic (4% O_2_) conditions in the presence of ApoAI. LXR agonist (indicated) was added to stimulate *ABCA1* expression. Values shown are means ± SD (n = 12 images per condition). One-way ANOVA with Tukey’s post hoc test. * *p* < 0.05. **** *p* < 0.0001.

**Table 1 ijms-23-03194-t001:** Genotypes of AMD-associated polymorphisms in all used iPSC lines.

Cell Line	*ABCA1* rs1883025 (Risk: C)	*ABCA1* rs2740488 (Risk: A)	*CFH* Y402H rs1061170 (Risk: C)	*ARMS2* A69S rs10490924 (Risk: T)	*C3* R102G rs2230199 (Risk: G)	*CFH* I62V rs800292 (Protective: A)	*CFHR3/1 del* rs12144939 (Protective: T)
A18945 (Thermo Fisher Scientific)	TC	CA	TT	GT	GG	GA	GG
IPS19-00104 (Decreased *ABCA1* risk)	TT	CC	TT	GT	GC	GG	GT
IPS19-00105 (Decreased *ABCA1* risk)	TT	CC	CT	GT	GC	GA	GG
IPS19-00106 (Decreased *ABCA1* risk)	TT	CC	CT	GT	GC	GG	GG
IPS19-00096 (Increased *ABCA1* risk)	CC	AA	CT	TT	GC	GG	GT
IPS19-00097 (Increased *ABCA1* risk)	CC	AA	TT	GG	GC	GA	GG
IPS19-00013 (Increased *ABCA1* risk)	CC	AA	CC	GG	GG	GG	GG

**Table 2 ijms-23-03194-t002:** Primer pairs for the amplification of cDNA derived from humans (h) or mouse (m).

Gene Name	Forward Primer (5′ to 3′)	Reverse Primer (5′ to 3′)
*hABCA1*	GGTCATGGCTGAGGTGAACA	TGGTCATTGTCCCTGCTGTC
*mAbca1*	GCGTGAAGCCTGTCATCTAC	CATGAGAGGAGTGATCGACC
*hABCG1*	CTCCTGTTCTCGGGGTTCTT	CCCTTCGAACCCATACCTGAC
*hACTB*	CCTGGCACCCAGCACAAT	GGGCCGGACTCGTCATAC
*mActb*	CAACGGCTCCGGCATGTGC	CTCTTGCTCTGGGCCTCG
*hADM*	ATCACTCTCTTAGCAGGGTCT	CCACTTATTCCACTTCTTTCG
*hBEST1*	CTTTATGGGCTCCACCTTCA	CAGTAGTTTGGTCCTTGAGTTTG
*hNR1H3*	TGCCCCATGGACACCTACA	TCTTGCCGCTTCAGTTTCTTC
*hOTX2*	AGTCGAGGGTGCAGGTATGG	TTTGACCTCCATTCTGCTGTTG
*hPDK1*	CACGCTGGGTAATGAGGATT	GGAGGTCTCAACACGAGGT
*hRLBP1*	GCTGCTGGAGAATGAGGAAAC	TGGCTGGTGGATGAAGTGG
*hRPE65*	TGACAAGGCTGACACAGGCA	CAAAGATGGGTTCTGATGGGTATG
*hRPL28*	GCAATTCCTTCCGCTACAAC	TGTTCTTGCGGATCATGTGT
*hSREBP-1C*	GGAGGGGTAGGGCCAACG	CATGTCTTCGAAAGTGCAATCC

## Data Availability

The data presented in this study are available on request from the corresponding author.

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
