# Peer review of "Regulation of ABCA1 by AMD-Associated Genetic Variants and Hypoxia in iPSC-RPE"

_ijms, 2022, doi:10.3390/ijms23063194_

Round 1

Reviewer 1 Report

The authors used IPSCs - derived iRPE cells from patients carrying the ABCA1 rs1883025 (C) and rs2740488 (A) genotype associated with increased risk of AMD. They showed a reduced efficiency of cholesterol efflux from the RPE in these cells. They showed that hypoxia reduced ABCA1 expression and increased intracellular lipid accumulation, in cells but also in mice, and that stimulation of ABCA1 transcription by an LXR agonist ameliorates cholesterol efflux in RPE cells. This finding highlights a possible treatment to correct efflux and intracellular accumulation of lipids in AMD patients.

This is a high-quality manuscript that brings new insights in the AMD pathophysiological mechanisms.

I have only three remarks:

In figure 2, the authors analyzed iRPE cells from patients with decreased and with increased risk, but I wondered what was the basal level in iRPE controls, for instance A19045 cell line harbouring a heterozygousABCA1 genotype? Why didn't you test them?

Line 228 "...promoter contains potential HIF1 binding elements": Can you define " hypoxia-inducible factor 1" before line 232?  

Same line 184, for readers unfamiliar with cholesterol investigation, can you define what is " BODIPY-cholesterol"?

Reviewer 2 Report

Present study explores the regulation of ABCA1 by AMD-associated genetic variants and hypoxia in iPSC-RPE. Authors used human iPSC derived RPE (iRPE) cells to investigate the influence of genetic variants and hypoxia on ABCA1 expression and function, which might be associated with the health status of RPE through intracellular lipid accumulation and in turn, lead to retinal degeneration and vision impairment. It is an interesting work but there are some concerns which needs to be addressed.

Comment 1: Authors are advised to include data for iPSC to RPE differentiation efficiency in the supplement section.

Comment 2: Authors are also advised to include pluripotency data for iPSC in the supplement section.

Comment 3: Did author analyzed off target genes for the gRNA used for generation of Exon 14 and Exon 40 knockouts.

Comment 4: Why authors selected to make Exon 14 and Exon 40 KO but most of the patient specific mutations are in intron between 3 and 4 exons.

Comment 5: Better representative images for Nile red staining can be used along with higher magnification.

Comment 6. Does authors have any explanation why there is increased ABCA1 expression after differentiation but not in undifferentiated cells.

Comment 7. For all the data authors are advised to include control line data with decreased risk and increased risk data to show comparative analysis.

Comment 8. For figure 3, authors are advised to use better representative blot.

Comment 9. What happen to gene expression and cholesterol efflux in KO lines under normoxia and hypoxia conditions?

Comment 10. Why authors think hypoxia alters ABCA1 expression? Authors can include explanation in the discussion section.

Round 2

Reviewer 2 Report

Authors have included the suggestions in he revised manuscript and this paper can be accepted in the present revised form.

This manuscript is a resubmission of an earlier submission. The following is a list of the peer review reports and author responses from that submission.

Round 1

Reviewer 1 Report

The manuscript by Peters et al. describes studies of the regulation and function of the cholesterol transporter, ABCA1 in iPSC derived RPE cells (iRPE) from patients with ABCA1 genotypes associated with AMD risk or protection as well as the effect of hypoxia on these iRPE cell lines and in vivo in mice.

Figure 1 depicts a plethora of rigorous data characterizing the ABCA1 ko iRPE cells and showing that these have reduced cholesterol efflux and increased accumulation of intracellular lipids. ABCG1 transcripts were significantly elevated in the human ko line but did not compensate for loss of ABCA1 as this group had previously shown in vivo in mouse RPE (PMID: 30864945).

Next, to interrogate the effect ABCA1 AMD risk and protective polymorphisms on ABCA1 expression and function in human RPE they generated iRPE from 6 AMD patients expressing the risk associated (n=3) or decreased risk associated (n=3) SNPs.

The iRPE lines were genotyped for ABCA1, CFH Y402H, ARMS2 A69S and C3 R102G AMD risk alleles. The 3 iRPE lines homozygous for decreased ABCA1 risk were fairly uniformly heterozygous at the other 3 loci. The 3 lines homozygous for increased ABCA1 risk are more heterogeneous. Based on the new paper by Hageman and collaborators (PMID: 34563268) showing that (1) the CFH-CFHR5 (on chromosome 1q32) and ARMS2/HTRA1 (on 10q26) loci account for the majority of genetic susceptibility for AMD; and (2) the 1q32 haplotypes (CFH I62V and a CFHR3/1 deletion tagging SNPs) mitigate the risk for AMD associated with both the CFH-CFHR5 and ARMS2/HTRA1 loci it might be prudent to genotype for these 2 protective haplotypes as well.

An unanticipated finding based the function of ABCA1 was that iRPE with the risk-associated ABCA1 expressed ABCA1 at higher levels but still had decreased cholesterol efflux.

This surprising since they also showed that hypoxia reduces ABCA1 expression with decreased cholesterol efflux and a large elevation in intracellular lipid levels. This could be reversed with an LXR agonist which increased ABCA1expression, increased efflux, and somewhat reduced intracellular lipid accumulation.

Overall, the findings are important to the vision community and beyond. As concluded and succinctly stated by the authors their study ‘supports the hypothesis that enhancing ABCA1-mediated efflux in RPE cells, e.g., by using an LXR agonist, could be a feasible treatment option to normalize efflux and intracellular accumulation of lipids in AMD patients.’  

Reviewer 2 Report

In this manuscript from the Grimm Lab, the authors describe experiments continuing their work on the role of reverse cholesterol export, specifically ABCA1, in RPE cells and its relevance to AMD. The main RPE cell culture system used by the authors was iPSC-derived RPE, which is an advantageous model to use since it retains many RPE characteristics (i.e. hexagonal shape, pigmentation, and gene expression as shown by the authors in Figure 1.) Using iPSC-derived RPE, the authors found differences in ABCA1 expression and function under normoxic and hypoxic conditions that are associated with AMD-associated SNPS within the ABCA1 gene. Thus, the authors conclude AMD-associated genetic variants and hypoxia can modulate ABCA1 expression and function within human RPE cells.

Major Note:

The manuscript is well-written and data is well-presented but this reviewer cautions the authors on their interpretation of their experiments on the iPSC-derived RPE cultures from patients with and without the AMD-associated ABCA1 SNPs. These cell cultures could either have variation in genes containing other AMD-associated SNPs such as APOE or variation in genes that could impinge on lipid handling within RPE cells. The best way to definitively show these AMD-associated SNPs in ABCA1 causes the increased ABCA1 expression and more cholesterol efflux when treated with direct BODIPY-cholesterol is by introducing the AMD-associated SNPs in ABCA1 through CRISPR-Cas9 technology into cells without these variants, differentiate them into RPE cells, and evaluate if similar results are obtained with those cultures. Thus, this reviewer recommends the authors use ‘associate’ rather than ‘affect’ when describing these results in their manuscript.

Other Minor Notes:

The iPSC-derived RPE cultures used in this study were grown on Matrigel-coated dishes and not transwell plates which would allow assessments of apical and basal cholesterol efflux. Presumably, the cholesterol efflux measured in this manuscript could be primarily from the apical side. Is there a reason why these cultures were not grown on transwell plates? This should be addressed in the discussion.

Also, the authors conclude that ABCG1 does not play a major role in cholesterol efflux in human RPE because it is not compensating for the loss of ABCA1. Furthermore, RPE-specific ABCG1 knockout mice do not have a phenotype which may be explained by the presence of functional ABCA1. Is it possible that ABCG1 could be primarily on the basal side of the RPE where it exerts its function? Thus, the increase in ABCG1 wouldn’t have an effect in this study since it is focused on apical cholesterol efflux? Have the authors generated ABCG1 knockout iPSC-derived RPE and evaluated their cholesterol efflux capacity? It would be interesting to specifically examined the apical and basal cholesterol efflux capacity with those cultures…

There are no data included in Figure 3 on untreated iPSC-derived RPE cultures. This should be included.

Reviewer 3 Report

This paper studied the regulation and function of ABCA1 in iRPE cells from patients with different polymorphisms, and identified hypoxia as a causative environmental condition to regulate ABCA1 expression. The study strengthened the role of ABCA1 in promoting cholesterol efflux from the RPE, consistent with previous published data, and supported therapeutic avenues aimed at increasing ABCA1-dependendent cholesterol efflux as a treatment for dry AMD. However, the mechanism of the genetic contributions of SNPs near ABCA1 to AMD risk remain unclear. The authors found that alleles linked to increased risk for developing AMD were associated with elevated ABCA1 expression in the iPSC/RPE, but also with decreased cholesterol efflux capacity. The elevated ABCA1 expression on cells with risk alleles is opposite of what the same group reported in patient-derived lymphoblastoid cells (Storti et al. 2019). These conflicting and confounding data were not explained. The mechanism mediating the effects of hypoxia on ABCA1 expression and cholesterol/lipid accumulation were not determined. Finally, with three samples in each group the authors were only able to generate marginally conclusive data, and they are generally overinterpreted. I suggest the authors increase the sample size and repeat the key experiments.

Supplemental figures were not able to be located for review.

References need to be added at lane 78, lane 364.

Figure 1A: scale bars are required for all images.

Figure 2: For figure 2B to 2E, the author stated that the ABCA1 relative expression were normalized to RPL28 for mRNA and Actin for western blot. In addition to this, was the data also normalized to decreased risk groups? Please clarify. Figure 2I, please indicate whether the images represent increased or decreased risk groups.

Figure 4: With the small sample sizes and high standard deviation, the authors are overinterpreting their data. I suggest the author to repeat this experiment with increased sample size to obtain a clear conclusion.

Figure 5: D-E: The effect of hypoxia is very minimal. Also, there is no control for LXR alone. In addition, it would be more convincing if cholesterol efflux and intracellular lipid accumulation are also investigated in vivofrom mice in Figure 5D.

5G: n=12, is it mean 12 images were counted/quantified? Please clarify.

Round 2

Reviewer 3 Report

Although the authors have provided comments/explanations in response to my review, and made minor changes to the text, I don't see any significant improvement to the manuscript. I maintain that the experiments are underpowered and with marginal data, and that it is not possible to draw conclusions from the results unless they are supported by additional more rigorous studies.